# Assessment of environmental risk factors for blastomycosis during a large outbreak at a Michigan paper mill

Allyson W. O'Connor[1], Ju-Hyeong Park[1], Marcia L. Stanton[1], Xiaoming Liang[1], Dallas Shi[2,3]*, Perri C. Callaway[1,4], R. Reid Harvey[1], Ryan LeBouf[1], Rachel L. Bailey[1], Ian Hennessee[4,5], Mitsuru Toda[5], Jennifer Meece[6], Alana Sterkel[7], Suzanne Dargle[7], Olivia Bree[7], Jeremy Olstadt[6], Rebecca Reik[8], Mary Grace Stobierski[8], Michael Snyder[9], Robert Yin[9], Marie A. de Perio[10], Stella Hines[1], Jean Cox-Ganser[1], on behalf of the Michigan Blastomycosis Outbreak Investigation Team¶

1 Respiratory Health Division, National Institute for Occupational Safety and Health, Centers for Disease Control and Prevention, Morgantown, West Virginia, United States of America, 2 Division of Field Studies and Engineering, National Institute for Occupational Safety and Health, Centers for Disease Control and Prevention, Cincinnati, Ohio, United States of America, 3 Epidemic Intelligence Service, Centers for Disease Control and Prevention, Atlanta, Georgia, United States of America, 4 Laboratory Leadership Service, Centers for Disease Control and Prevention, Atlanta, Georgia, United States of America, 5 Mycotic Diseases Branch, Centers for Disease Control and Prevention, Atlanta, Georgia, United States of America, 6 Marshfield Clinic Research Institute, Marshfield, Wisconsin, United States of America, 7 University of Wisconsin-Madison and Wisconsin State Laboratory of Hygiene, Madison, Wisconsin, United States of America, 8 Michigan Department of Health and Human Services, Lansing, Michigan, United States of America, 9 Public Health Delta & Menominee Counties, Escanaba, Michigan, United States of America, 10 Office of the Director, National Institute for Occupational Safety and Health, Centers for Disease Control and Prevention, Cincinnati, Ohio, United States of America

¶ Membership of the Michigan Blastomycosis Outbreak Investigation Team is provided in the Acknowledgments.
* rhq9@cdc.gov

## Abstract

### Background

Blastomycosis is a rare, potentially fatal fungal infection caused by inhalation of *Blastomyces* spores, typically acquired outdoors in the midwestern and eastern United States. In 2023, the largest recorded U.S. blastomycosis outbreak occurred among workers at a paper mill in Michigan's Upper Peninsula. Few data exist on occupational risk factors or indoor exposure to *Blastomyces*, limiting prevention efforts.

### Objectives

We assessed workplace environments and conditions associated with blastomycosis risk through a cross-sectional medical survey and environmental sampling.

### Methods

During April 22–28, 2023, we conducted a voluntary medical survey, including a work and health questionnaire and urine antigen testing, for 603 workers out of

**Data availability statement:** The data that support the findings of this evaluation are not publicly available due to privacy requirements. While there are hundreds of workers in the dataset, the number of workers in various work locations as well as the number who responded to certain questions is sufficiently small to risk worker identification if released. The data collected for this Health Hazard Evaluation is protected under The Privacy Act and, as such, is prohibited from disclosure "…except pursuant to a written request by, or with the prior written consent of, the individual to whom the record pertains [subject to 12 exceptions]." 5. U.S.C. §552a(b). https://www.govinfo.gov/content/pkg/USCODE-2018-title5/pdf/USCODE-2018-title5-partI-chap5-subchapII-sec552a.pdf. NIOSH and its employees could be subject to civil and criminal penalties for unauthorized disclosures of Privacy Act-protected information. 5 U.S.C. §§ 552a(g), 552a(i). Limited data may be available upon request to the Centers for Disease Control and Prevention Freedom of Information Act (FOIA) Office at https://www.cdc.gov/foia/index.html.

**Funding:** The author(s) received no specific funding for this work.

**Competing interests:** The authors have declared that no competing interests exist.

approximately 1,000 at the mill. We compared worker characteristics, work locations, and environmental exposures by blastomycosis case status and modeled disease risk using Poisson regression. We tested 533 environmental samples of outdoor soil, indoor surface dust, and raw materials for *Blastomyces* using polymerase chain reaction and culture-based methods.

## Results

Twenty percent of workers were classified as blastomycosis cases based on positive urine antigen testing during the survey, self-reported provider diagnoses, or confirmed or probable case status reported by state or local health departments. Prevalence was highest among workers in paper machine line #1 (27%) and maintenance areas (25%). Adjusted analyses indicated a 40% [Prevalence Ratio (PR): 1.40; 95% confidence interval (CI): 1.00, 1.95] and 53% (PR: 1.53; 95% CI: 1.04, 2.25) higher risk for workers in these locations, respectively, compared to workers working elsewhere. Working in both locations doubled blastomycosis risk. Daily exposure to indoor pooling water was associated with a nearly two-fold higher prevalence of blastomycosis (PR: 1.79; 95% CI: 1.25, 2.57). All indoor and outdoor environmental samples were negative for *Blastomyces*.

## Discussion

Blastomycosis was associated with specific indoor work locations and environmental conditions, suggesting the potential for occupational exposure to *Blastomyces* in indoor industrial settings. These findings may guide future outbreak investigations and occupational prevention strategies.

## Introduction

During 2022–2023, the largest documented outbreak of blastomycosis, a rare but potentially fatal fungal infection, occurred at a paper mill in the Upper Peninsula of Michigan [1]. This infection typically arises from inhaling spores of the *Blastomyces* fungus. *Blastomyces* naturally inhabits the environment and can cause respiratory and systemic symptoms that mimic other illnesses, potentially leading to diagnostic delays and severe pulmonary complications [2,3].

Disease surveillance in the United States (U.S.) captures approximately 200–300 cases of blastomycosis each year [4], primarily consisting of sporadic cases, although clusters and outbreaks occur intermittently. Cases have been linked to outdoor activities and proximity to water sources, such as lakes, ponds, and rivers, especially in the midwestern and eastern United States where the fungus is thought to be endemic [5]. Additionally, certain outdoor occupations like natural resource management and construction work that disturb soil have been linked to the disease [6].

In late February 2023, a cluster of blastomycosis cases among mill workers was identified by the local health department in Delta County, Michigan—an area where

*Blastomyces* is endemic and sporadic cases occur annually. This detection prompted a coordinated multi-agency public health response involving mill management, workers and their representatives, and local, state, and federal health officials. This outbreak of blastomycosis among paper mill workers was the first instance linked to an industrial, largely indoor worksite [1].

The National Institute for Occupational Safety and Health (NIOSH) at the Centers for Disease Control and Prevention (CDC), conducted a workplace evaluation at the mill [7] to investigate *Blastomyces* exposure sources, identify undiagnosed cases, and recommend measures to prevent further illness. Since blastomycosis is environmentally acquired, assessing work processes, locations, and environmental conditions was crucial for understanding exposure risks.

Paper milling involves refining raw wood materials into paper through water and chemically intensive processes [8]. The process begins with chipping the wood, which is then converted into pulp through mechanical or chemical breakdown. Machines press and roll the wet pulp into dry paper rolls, which are cut to size for printing. Observational studies have linked soft paper dust from paper milling to respiratory symptoms and impairments [9–11] and associated sulfite and sulfate chemicals used in pulp processing with increased mortality [12].

The mill site, situated on forested land along a riverway, features a large complex of mill buildings, stacks of trees and wood chip piles for pulp processing, settling ponds for treating chemical effluent before it is released into the river, parking areas on either side of the river with footbridge access to the mill, and a network of unpaved roads. The mill property covers more than 2,000 acres, and most paper processing activities take place indoors after the wood is chipped.

Although isolating *Blastomyces* in the environment is challenging and its ecological niche remains unclear [13], it is thought to grow in moist, organic-rich soil, raising concerns about potential exposure from the woodyard and riverway. Blastomycosis is usually contracted through outdoor exposure. Indoor transmission is rarely reported [14]; thus, investigating how the predominantly indoor workforce was exposed to *Blastomyces* was a key focus of the investigation [13]. After an initial site visit to the mill in March 2023, we conducted a follow up medical and environmental survey to evaluate work locations and environmental conditions at the mill associated with blastomycosis. We also investigated potential indoor and outdoor sources of *Blastomyces* exposure through environmental sampling on mill grounds. Our aim was to assess work environments and conditions associated with blastomycosis risk and identify potential environmental sources through a cross-sectional survey and environmental sampling. Findings from this investigation may help inform future outbreak investigations and guide prevention strategies in occupational settings where endemic mycoses may be introduced indoors.

## Methods

### Medical and environmental survey

During April 22–28, 2023, all mill employees and contractors (approximately 1,000 workers) were invited to participate in a voluntary medical survey as part of NIOSH's Health Hazard Evaluation (HHE) requested by mill management to investigate the blastomycosis outbreak. The survey, conducted near the mill, included a work and health questionnaire and *Blastomyces* antigen testing through urine sampling. No serologic or skin testing was conducted as part of this investigation to identify asymptomatic infection. Skin tests are no longer widely available, and most serologic assays have low sensitivity and limited specificity for blastomycosis [15]. Although a newer enzyme immunoassay for antibody detection offers improved performance, its clinical utility in distinguishing recent from historical infections remains uncertain. Blood collection also introduces logistical, biosafety, and consent-related challenges that make urine antigen testing a more feasible approach for large-scale field investigations.

Trained interviewers administered a structured questionnaire that captured participants' self-reported information on work history and locations, environmental conditions, medical history, health symptoms, recent respiratory illnesses, blastomycosis-related experiences, and non-work-related activities, including potential recreational or community-based exposures to *Blastomyces* (e.g., hiking, gardening, hunting, or time spent along the river). The questionnaire also included

items on use of personal protective equipment (PPE), including respiratory protection, gloves, and other safety gear during work activities. This report focuses on environmental risk factors for blastomycosis encountered by workers while at the mill, since all reported cases were among mill workers. Clinical characteristics of affected workers were previously reported [1,16].

## Work locations

During questionnaire completion, survey administrators used mill maps to help workers identify their primary work location, up to three secondary work locations, personal offices, and breakrooms/lunchrooms/shacks for up to three of their most recent jobs held at the mill since October 1, 2022—three months before the earliest known case. This timeframe was selected to capture potential exposure locations based on the long latency period of blastomycosis, which ranges from two weeks to three months [3,17].

Work locations were categorized by mill processes, with all papermaking machines grouped into three distinct paper machine lines. Maintenance areas were grouped together for analysis, though they were physically spread throughout the mill and included spaces in the basement of paper machine line #1. Most workers entered the mill through turnstiles located near the administrative buildings, which were adjacent to a river. Workers accessed the turnstiles either by crossing a footbridge from Parking Lot A or by walking from Parking Lot B near the shipping bays. Contractors parked separately in a designated lot north of the mill. Many workers passed through the administrative offices on their way to their primary work areas, regardless of their final destination within the mill.

## Environmental conditions

Workers were surveyed about their frequency of exposure to seven specific environmental conditions around the mill since October 1, 2022. These conditions included damp or wet mulch, dry mulch or sawdust, wet soil, standing pools of water indoors, standing pools of water outdoors, wind gusts indoors, and visible mold on indoor surfaces. Workers also identified the primary work location where they encountered these conditions. Investigators also conducted qualitative assessments of the mill's ventilation systems, including heating, ventilation, and air conditioning (HVAC) operation, airflow direction, visible damage to ductwork, and potential for cross-contamination between work areas.

## Case definition

Blastomycosis cases among mill workers were identified through multiple sources. Workers were considered cases if they (1) had a positive urine antigen test result from the NIOSH survey, (2) self-reported a provider-diagnosed case during the survey interview, and/or (3) were reported as a confirmed or probable case by Public Health Delta & Menominee Counties (PHDM) or the Michigan Department of Health and Human Services (MDHHS). Confirmed cases met both clinical and laboratory criteria for blastomycosis; probable cases met clinical criteria with supportive (but not confirmatory) laboratory evidence, as defined by the Council of State and Territorial Epidemiologists [18]. MDHHS defined outbreak cases as persons with illness onset on or after December 1, 2022, who met CSTE criteria and had resided in or spent at least 40 hours in Delta County since September 1, 2022 [1,18]. Case information from PHDM and MDHHS was used to supplement NIOSH survey data, including missing illness onset dates.

## Data analysis

We analyzed characteristics of workers in the NIOSH medical survey by blastomycosis case status using Wilcoxon rank sum tests for continuous variables and Pearson's chi-squared tests for categorical variables ($\alpha = 0.05$ for significant associations; $\alpha = 0.10$ for marginally significant associations). Key characteristics included reported environmental exposures, daily turnstile usage, passage through administrative offices, and parking lot use. We mapped the prevalence of blastomycosis by work location and visualized illness onset dates using kernel density plots.

We used Poisson regression models with robust standard errors ($\alpha = 0.05$) to assess associations between blastomycosis case status, reported work locations, and environmental conditions. Directed acyclic graphs (DAGs) were developed to conceptualize causal relationships and identify adjustment variables (S1 Fig). For work location analyses, we estimated prevalence ratios (PRs) as the prevalence of blastomycosis among workers who reported working in specific locations (as a primary or secondary work area, office, or breakroom/lunchroom/shack) compared to the prevalence of disease for those who did not work there, adjusting for tenure at the mill and sex. For primary work locations, we used effect coding to compare the prevalence of disease in each area to the overall prevalence of blastomycosis across the mill.

For environmental conditions, we estimated PRs for daily exposure compared to less frequent or no exposure (i.e., some days, rarely, or never) to conditions, adjusting for mill tenure, sex, and primary work location. Environmental conditions were modeled individually and for multiple conditions (e.g., exposure to indoor pooling water and visible mold indoors). Analyses were conducted using SAS® software version 9.4 (SAS Institute Inc., Cary, NC), JMP® version 16.1 (SAS Institute Inc., Cary, NC) [19], and R (version 4.5.1; https://www.r-project.org/). DAGs were created using DAGitty version 3.1. (www.dagitty.net; developed by Johannes Textor, Radboud University Medical Center, Nijmegen, The Netherlands) [20].

## Environmental sampling and analyses

NIOSH staff performed environmental sampling indoors and outdoors on mill property during April 24–28 and August 1–2, 2023, to investigate potential environmental sources of *Blastomyces* and assess conditions that may support fungal persistence. We conducted the main indoor and outdoor sampling during the medical survey period in April. Additional outdoor sampling occurred in August, coinciding with the construction of a bridge within the mill property. Indoor and outdoor sampling locations were selected based on three key criteria: where workers with known blastomycosis cases worked, their proximity to suspected environmental reservoirs (e.g., riverbanks, ponds, wood chip piles), and evidence of water intrusion. We prioritized the wood chip storage areas due to their high organic content and contact with outdoor soil, which may support fungal growth. For indoor locations, HVAC filters, duct liners, and accumulated surface dust were targeted to assess whether spores may have been entrained, redistributed through ventilation, or settled in shared workspaces. A total of 533 environmental samples were collected, including outdoor soils (n = 113), raw wood materials (n = 9), indoor surface dust (n = 344), and HVAC filters, duct liners, and pooled water (n = 67). These sample types reflect potential environmental reservoirs or transport pathways for *Blastomyces*, with an emphasis on both entry from the outdoor environment and possible persistence or movement within indoor work areas.

Surface dust samples were collected by wiping accumulated dust on elevated surfaces indoors, including the top of cabinets, desktops, machine surfaces, computer surfaces, HVAC surfaces, and the top of door frames. Surfaces were wiped using a sterile technique with an electrostatic cloth measuring approximately 2 x 2 inches; nitrile gloves were changed for each sample to prevent cross-contamination. The dust-laden cloths were then placed into sterile, capped 50-milliliter (mL) conical tubes. Approximately 45 mL of soil and bulk material from the banks of a river near the mill were collected into 50 mL conical tubes using a sterile spatula. Used filters and duct liners available from HVAC systems were also collected.

Environmental samples were transported to a Biosafety Level 3 laboratory at the Marshfield Clinic Research Institute (MCRI) and stored at −80°C (soil, water, and surface dust samples) or refrigerated (air filter and cut liner samples) until analysis. MCRI processed the samples for *Blastomyces* DNA analysis and prepared aliquots for culturing at the Wisconsin State Laboratory of Hygiene (WSLH) at the University of Wisconsin-Madison. Soil, electrostatic cloth, and filter samples were prepared and analyzed in parallel at both MCRI and WSLH. At MCRI, fungal DNA was extracted with Qiagen® (Germantown, MD) and Roche kits (Indianapolis, IN).

Polymerase chain reactions with the following primer targets were run on the isolated DNA samples: *Blastomyces* adhesion-1 (*BAD-1*) gene, which is specific for *Blastomyces,* and the *ITS3/4* gene, as a positive control found in all fungi, including *Blastomyces*. Reaction products were then assessed by gel electrophoresis for presence of the target genes. WSLH cultured all samples using four different media [Sabouraud's dextrose agar (SAB); brain heart infusion agar (BHI)

with and without chloramphenicol and gentamicin; and BHI with chloramphenicol, gentamicin, and cycloheximide] to grow fungi at 39°C for 8 weeks. This elevated temperature selected for yeast phase growth of *Blastomyces* and suppressed growth of mold contaminates. Yeast-like growth was then isolated by sub-culturing onto antimicrobial-free BHI and SAB. Pure isolates were evaluated using microscopy and tested with Matrix-Assisted Laser Desorption/Ionization Time-of-Flight mass spectrometry (MALDI-ToF) to exclude identifiable non-*Blastomyces* yeast species. Although MALDI-ToF does not have sufficient reference material in its database of reference spectra to identify *Blastomyces*, it can successfully identify many types of bacteria and yeast to help reduce the number of isolates needing PCR (Polymerase Chain Reaction) level identification. Isolates that were unable to be identified by MALDI-ToF and visually resembled *Blastomyces* broad-based budding yeasts were subsequently reanalyzed by PCR for identification. Additional details about the laboratory procedures and experiments to determine the limit of detection for the environmental samples will be reported separately.

### Ethical review and participant consent

This activity was reviewed by CDC and conducted in compliance with applicable federal law and CDC policy (45 C.F.R. part 46.102(l)(2); 21 C.F.R. part 56; 42 U.S.C. §241(d); 5 U.S.C. §552a; 44 U.S.C. §3501 et seq.). The study was determined not to be human subjects research but to be public health practice as part of the NIOSH Health Hazard Evaluation Program and thus was not subject to institutional review board review. All participants provided written informed consent before participation in the medical survey. The informed consent included information about the purpose and duration of the medical survey, what participants would be asked to do, and the benefits and minimal risks involved, as well as that personal information collected was confidential and would be protected to the extent allowed by law, participation was voluntary, and participants could withdraw from the evaluation at any time, for any reason.

## Results

Among the 608 workers who participated in the NIOSH medical survey, complete questionnaire data were available for 603 workers, and urine antigen testing results were available for 573 workers. Among survey participants, 120 (20%) workers were classified as having blastomycosis cases. The median age was 47 years (range: 19–73), and the median tenure at the mill was 8 years (range: 0–52). Most participants were male (82%). Younger age (p = 0.04) and shorter tenure (p = 0.03) were significantly associated with blastomycosis case status (Table 1).

### Work location

Overall, blastomycosis prevalence was highest in maintenance areas (27%) and paper machine line #1 (25%)—two of the most commonly reported work locations—followed by finishing and shipping (23%), the woodyard (21%), other paper machine lines (15–20%), and administrative offices (19%) (Fig 1). In adjusted analyses for all reported work locations, the prevalence of blastomycosis was 40% higher (PR: 1.40; 95% CI: 1.00, 1.95) among workers in paper machine line #1 and 53% higher (PR: 1.53; 95% CI: 1.04, 2.25) among those in maintenance areas compared to workers who did not report working in these locations from October 1, 2022 to the time of the survey in late April 2023 (Fig 2). Furthermore, working in both paper machine line #1 and the maintenance areas was associated with more than a two-fold greater risk of blastomycosis (Fig 3; PR: 2.17; 95% CI: 1.35, 3.49) compared to not working in either area.

The risk of blastomycosis did not vary significantly based on primary work location when compared to the overall prevalence at the mill (S1 Table). The case prevalence for each area of the mill, categorized by workers' primary work locations, is shown in S2 Fig, and illness onset dates by location are shown in S3 Fig.

The mean blastomycosis illness onset date was March 4, 2023, and the median date was March 7, 2023 (S3 Fig). The earliest onset cases occurred among workers primarily working in finishing and shipping, the maintenance areas, administrative offices, and the woodyard. Most cases at the mill had an illness onset date between January and April of 2023, regardless of primary work location.

**Table 1. Characteristics of workers who participated in the medical survey.**

| Characteristics | All workers n = 603[a] | Blastomycosis status | | p-value[b] |
|---|---|---|---|---|
| | | Case n = 120[a] | Non-Case n = 483[a] | |
| **Age, years, median (range)** | 47 (19–73) | 45 (20–67) | 47 (19–73) | 0.04 |
| **Sex, No. (Col. %)** | | | | |
| Male | 495 (82) | 100 (83) | 395 (82) | 0.69 |
| Female | 108 (18) | 20 (17) | 88 (18) | |
| **Mill tenure, years, median (range)** | 8 (0–52) | 7 (0–44) | 9 (0–52) | 0.03 |
| **Department, No. (Col. %)** | | | | |
| Administrative Offices | 93 (15) | 18 (15) | 75 (16) | 0.53 |
| Fiberline | 44 (7) | 12 (10) | 32 (7) | |
| Recovery and utilization | 38 (6) | --[c] | 35 (7) | |
| Maintenance and engineering | 140 (23) | 27 (23) | 113 (23) | |
| Paper machine | 189 (31) | 40 (33) | 149 (31) | |
| Finishing and shipping | 51 (9) | 12 (10) | 39 (8) | |
| Wood and coal yard | 33 (6) | 5 (4) | 28 (6) | |
| Other | 15 (3) | --[c] | 12 (2) | |

[a]This is the maximum number of respondents. Number of respondents for each characteristic varied due to missing data.

[b]Wilcoxon rank-sum tests were used for continuous variables and chi-square tests were used for categorical variables to test unadjusted differences by blastomycosis case status.

[c]Number and percentage omitted to avoid worker identification because of small numbers of responses (fewer than five) in at least one cell.

## Environmental conditions

Daily exposures to each environmental condition at the mill were detailed in Table 2, which presented unadjusted proportions and p-values by case status. Conditions that were significantly (p-values ≤ 0.05) or marginally significantly (p-values ≤ 0.10) associated with blastomycosis included indoor pooling water (27% among cases vs. 15% among non-cases, p = 0.002), visible mold indoors (28% vs. 20%, p = 0.04), gusts of wind indoors (40% vs. 32%, p = 0.09), and outdoor pooling water (12% vs. 7%, p = 0.10).

After adjusting for potential confounders (Fig 4), daily exposure to indoor pooling water remained significantly associated with blastomycosis (PR = 1.79, 95% CI: 1.25–2.57), whereas visible mold indoors (PR = 1.43, 95% CI: 0.99–2.08) and outdoor pooling water (PR = 1.82, 95% CI: 1.11–2.98) were marginally or significantly associated.

We also examined combinations of exposures (Fig 5A–C). Workers who reported daily exposure to both indoor and outdoor pooling water (Fig 5A) had more than twice the prevalence of blastomycosis (PR = 2.53; 95% CI: 1.42–4.49) compared to those not exposed to either. Similarly, workers exposed to both indoor pooling water and visible mold (Fig 5B) had a higher prevalence (PR = 2.10; 95% CI: 1.24–3.57), whereas visible mold alone was not associated with blastomycosis. Daily exposure to indoor wind (Fig 5C) was associated with blastomycosis only when not accompanied by indoor pooling water (PR = 1.51; 95% CI: 1.01–2.23). Workers who reported daily indoor pooling water exposure, but no wind exposure had a nearly three-fold higher prevalence of blastomycosis (PR = 2.74; 95% CI: 1.70–4.42).

Daily exposure to indoor pooling water, visible mold, and indoor wind gusts were reported most often in paper machine line #1. Indoor pooling water and visible mold indoors were reported second most often in paper machine line #4, whereas indoor wind gusts were reported second most often in the finishing and shipping area (S4–S6 Figs).

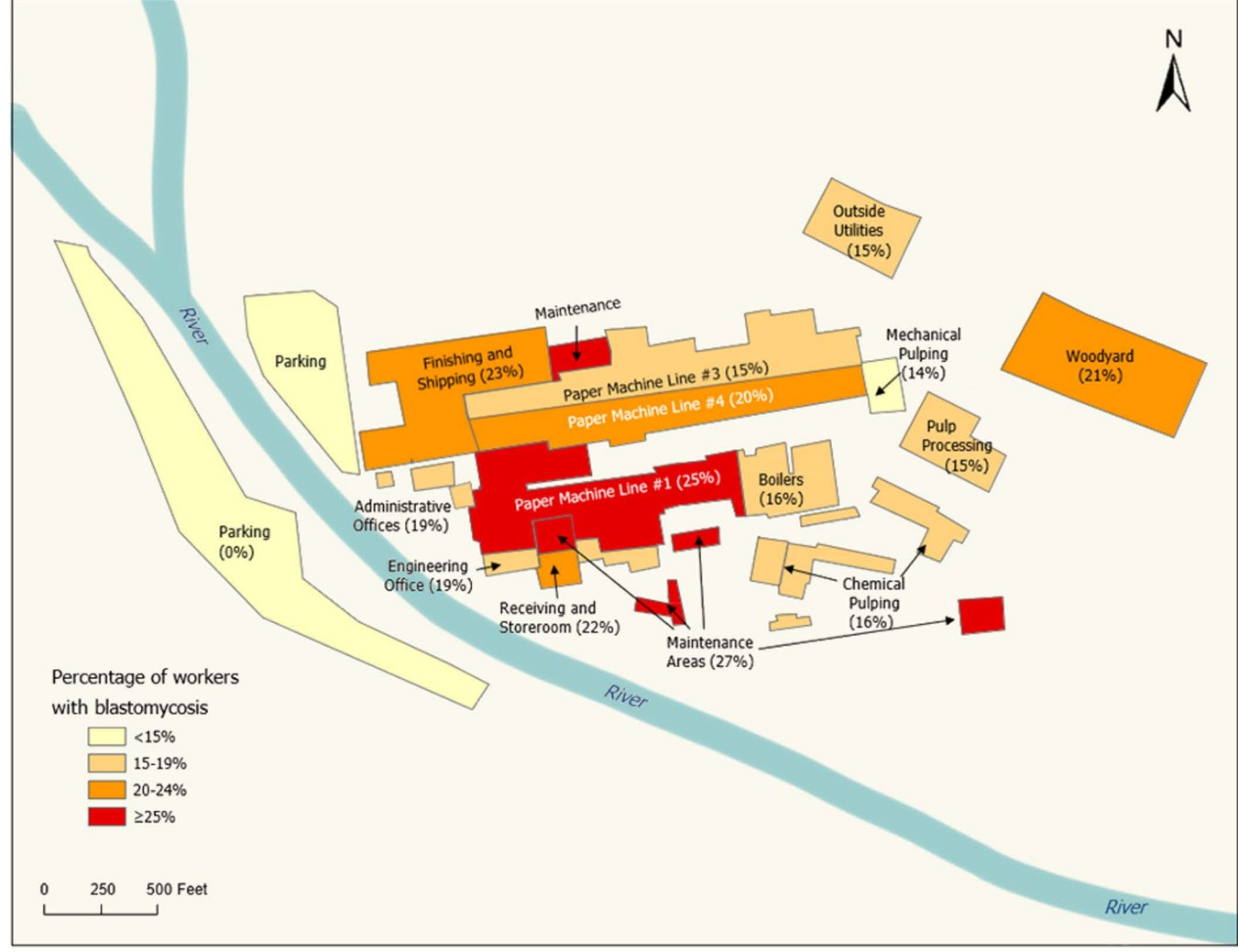

**Fig 1. Map of blastomycosis prevalence by mill area.** The prevalence of blastomycosis at the mill was evaluated for each process area based on workers' reported primary and secondary work locations, office spaces, and breakrooms/lunchrooms/shacks from October 2022–April 2023.

Investigators observed variable ventilation conditions across the mill, with some indoor areas exhibiting poor air-flow, standing water, or musty odors. HVAC systems showed signs of moisture accumulation and dust buildup. Survey responses indicated that most workers did not routinely wear respiratory protection; PPE use varied by department and task, with respiratory PPE primarily limited to specific job functions involving chemical exposures.

## Environmental sampling

We collected and analyzed 533 environmental samples across four major categories: indoor surface dust (n = 344), outdoor soil and bulk materials (n = 122), HVAC filters or duct liners (n = 66), and pooled water from the HVAC system (n = 1). A summary of these sample types and findings was provided in S2 Table.

All samples tested negative for *Blastomyces* using PCR targeting the *BAD-1* gene, regardless of sample type, preparation, or extraction method. To confirm PCR performance, we tested for general fungal DNA using primers targeting the internal transcribed spacer (*ITS3/4*) region of fungal rRNA genes. To assess for potential inhibition, we conducted

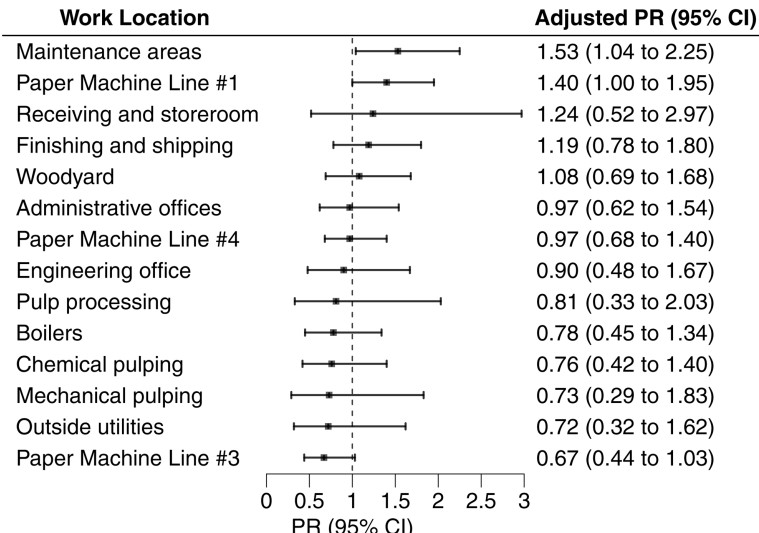

**Fig 2. Blastomycosis risk by work location.** Adjusted prevalence ratios (PR) and 95% confidence intervals (CI) estimated risk of blastomycosis for working versus not working in each location, including for workers' primary and secondary work locations, office spaces, and breakrooms/lunchrooms/shacks, from October 2022–April 2023. PRs were adjusted for mill tenure and sex.

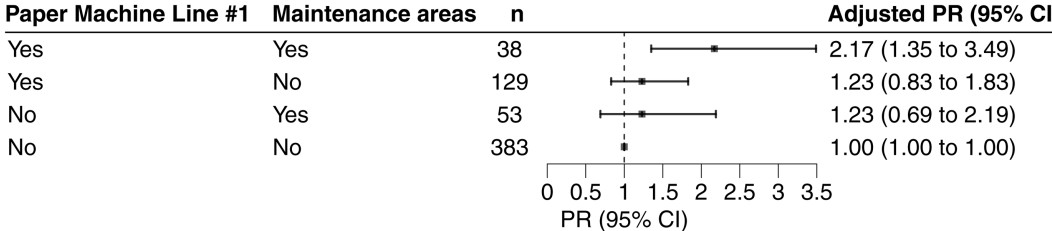

**Fig 3. Blastomycosis risk associated with Paper Machine Line #1 and the Maintenance areas.** Adjusted prevalence ratio (PR) and 95% confidence intervals (CI) estimated risk of blastomycosis for working in paper machine line #1 and/or the maintenance areas compared to not working in either of these mill locations from October 2022–April 2023. PRs were adjusted for mill tenure and sex; n indicates number of workers per location.

separate inhibition control experiments. General fungal DNA was detected in the majority of samples—specifically, in 100% of outdoor soil and bulk samples, 95% of HVAC filters, 62% of indoor dust samples, and in the single HVAC water sample.

All samples were also cultured on multiple types of selective media. General fungal growth was observed in most samples, including 95% of indoor dust samples, 93% of outdoor soil/bulk samples, and 25% of HVAC filter samples. Environmental molds, particularly *Aspergillus* species, rapidly overgrew most culture plates. However, yeast-like colonies with smooth, creamy morphology were occasionally observed, especially on brain heart infusion (BHI) agar supplemented with chloramphenicol, gentamicin, and cycloheximide, which limited mold overgrowth. Most of these yeast-like isolates were identified via MALDI-ToF as non-*Blastomyces* fungi or bacteria including *Pseudomonas*, *Candida*, or *Rhodotorula*. The remaining 22 non-identifiable isolates that resembled *Blastomyces* morphologically were tested using *BAD-1* PCR but were all negative. However, they were positive using *ITS3/4* primers, confirming they were fungi, but not *Blastomyces*.

**Table 2. Environmental exposures at the mill reported from October 2022–April 2023.**

| Exposures | All workers n = 603[a] | Blastomycosis status | | p-value[b] |
| --- | --- | --- | --- | --- |
| | | Case n = 120[a] | Non-Case n = 483[a] | |
| **Daily exposure (vs. < daily exposure) to environmental conditions, No. (%)** | | | | |
| Standing pools of water indoors on mill property | 102 (17) | 32 (27) | 70 (15) | 0.002 |
| Visible mold on indoor surfaces, walls, or ceilings | 128 (21) | 34 (28) | 94 (20) | 0.04 |
| Wind gusts indoors or blowing air while inside the mill | 201 (33) | 48 (40) | 153 (32) | 0.09 |
| Standing pools of water outdoors on mill property | 48 (8) | 14 (12) | 34 (7) | 0.10 |
| Damp or wet mulch or pulp | 128 (21) | 29 (24) | 99 (21) | 0.39 |
| Dry mulch or sawdust | 84 (14) | 14 (12) | 70 (15) | 0.41 |
| Wet soil | 38 (6) | 8 (7) | 30 (6) | 0.87 |
| **Passed through the front turnstile daily vs. not every day, No. (%)** | 564 (94) | 117 (98) | 447 (93) | 0.03 |
| **Walked through administrative offices daily vs. not every day, No. (%)** | 190 (32) | 40 (34) | 150 (31) | 0.64 |
| **Parking lot use, No. (%)** | | | | |
| Lots A and B | 320 (54) | 69 (58) | 251 (53) | 0.64 |
| Always lot A | 243 (41) | 44 (37) | 199 (42) | |
| Always lot B | 16 (3) | --[c] | --[c] | |
| Contractor lot | 16 (3) | --[c] | --[c] | |

[a]This is the maximum number of respondents for each exposure question among workers who participated in the NIOSH medical survey. Number of respondents varied due to missing data.

[b]Chi-square or Fisher's exact tests were used for categorical variables to test unadjusted differences by the case definition.

[c]No. (%) omitted to avoid worker identification from the small numbers of responses (n < 5).

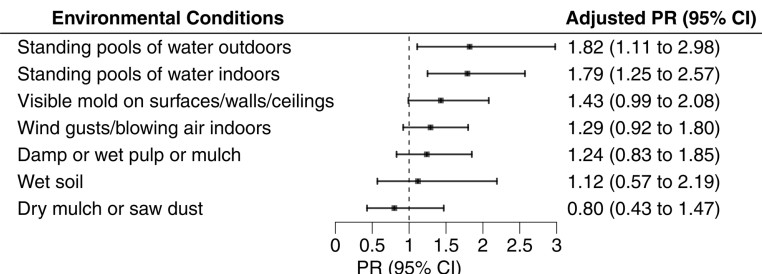

| Environmental Conditions | Adjusted PR (95% CI) |
| --- | --- |
| Standing pools of water outdoors | 1.82 (1.11 to 2.98) |
| Standing pools of water indoors | 1.79 (1.25 to 2.57) |
| Visible mold on surfaces/walls/ceilings | 1.43 (0.99 to 2.08) |
| Wind gusts/blowing air indoors | 1.29 (0.92 to 1.80) |
| Damp or wet pulp or mulch | 1.24 (0.83 to 1.85) |
| Wet soil | 1.12 (0.57 to 2.19) |
| Dry mulch or saw dust | 0.80 (0.43 to 1.47) |

**Fig 4. Blastomycosis risk associated with daily exposure to environmental conditions at the mill.** Adjusted prevalence ratios (PR) and 95% confidence intervals (CI) estimated risk of blastomycosis associated with daily versus less than daily exposure to environmental conditions at the mill from October 2022–April 2023. PRs were adjusted for mill tenure, sex, and primary work location.

These results indicate that fungi were widely present in the sampled environments—particularly in soil and indoor dust—but *Blastomyces* was either absent or present at levels below the detection limit of our PCR and culture methods.

## Discussion

We investigated a large outbreak of blastomycosis among paper mill workers in Michigan, representing the largest recorded outbreak of this rare fungal disease and the first known to occur in a predominantly indoor industrial setting. Although blastomycosis is endemic in this region and sporadic cases occur annually, the size and concentration of illnesses in a single workplace were unprecedented. Our findings suggest that exposure to *Blastomyces* spores may have

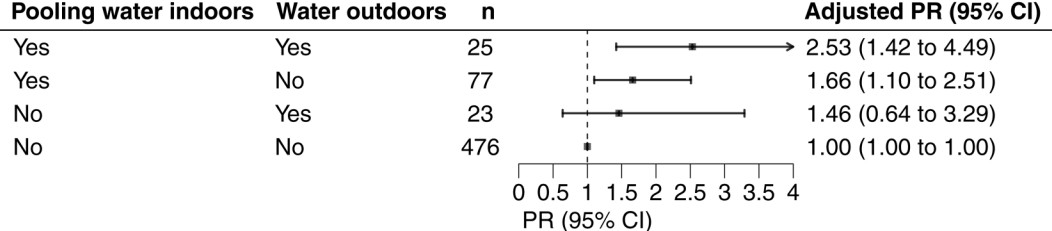

**A: Combination of indoor and outdoor pooling water**

| Pooling water indoors | Water outdoors | n | | Adjusted PR (95% CI) |
|---|---|---|---|---|
| Yes | Yes | 25 | | 2.53 (1.42 to 4.49) |
| Yes | No | 77 | | 1.66 (1.10 to 2.51) |
| No | Yes | 23 | | 1.46 (0.64 to 3.29) |
| No | No | 476 | | 1.00 (1.00 to 1.00) |

0 0.5 1 1.5 2 2.5 3 3.5 4
PR (95% CI)

**B: Combination of indoor pooling water and visible mold**

| Pooling water indoors | Visible mold | n | | Adjusted PR (95% CI) |
|---|---|---|---|---|
| Yes | Yes | 40 | | 2.10 (1.24 to 3.57) |
| Yes | No | 61 | | 1.95 (1.25 to 3.04) |
| No | Yes | 88 | | 1.47 (0.93 to 2.35) |
| No | No | 412 | | 1.00 (1.00 to 1.00) |

0 0.5 1 1.5 2 2.5 3 3.5 4
PR (95% CI)

**C: Combination of indoor pooling water and indoor wind**

| Pooling water indoors | Wind indoors | n | | Adjusted PR (95% CI) |
|---|---|---|---|---|
| Yes | Yes | 59 | | 1.63 (0.97 to 2.75) |
| Yes | No | 43 | | 2.74 (1.70 to 4.42) |
| No | Yes | 142 | | 1.51 (1.01 to 2.23) |
| No | No | 357 | | 1.00 (1.00 to 1.00) |

0 0.5 1 1.5 2 2.5 3 3.5 4
PR (95% CI)

**Fig 5. Blastomycosis risk associated with daily exposure to multiple environmental conditions.** Adjusted prevalence ratio (PR) and 95% confidence intervals (CI) estimated risk of blastomycosis for daily versus less than daily exposure from October 2022–April 2023 at the mill to: (A) pooling water indoors and pooling water outdoors, (B) pooling water indoors and visible mold indoors, and (C) pooling water indoors and wind indoors. PRs were adjusted for mill tenure, sex, and primary work location; n indicates number of workers reporting daily exposure to the conditions.

occurred indoors at the mill—an unusual setting for blastomycosis—and that certain work locations and environmental conditions were associated with elevated disease prevalence.

*Blastomyces* is known to inhabit moist, organic-rich environments, such as riverbanks, wooded areas, and decaying vegetation [5,21,22]. The mill is located adjacent to a river and includes large volumes of wood and organic material, potentially providing favorable conditions for fungal growth. Weather data from Delta County indicated that the winter preceding the outbreak (December 2022–February 2023) was milder than average, with higher-than-normal temperatures and below-average snowfall [23]. A mild winter prior to the outbreak may have further supported spore viability in the environment.

Sporadic cases of blastomycosis are reported annually in the Upper Peninsula of Michigan, but this outbreak was unusual in scope. Whole-genome sequencing of clinical isolates by the health department revealed close genetic relatedness between outbreak cases and a local canine isolate, suggesting the possibility of a local environmental source near the mill [16]. Findings from our investigation further support this conclusion: associations between blastomycosis and specific indoor environmental, such as visible mold, indoor pooling water, and gusts of wind indoors, support the possibility of exposure occurring within the indoor workplace, although we cannot determine this with certainty. Together, the epidemiologic and genomic findings are consistent with a plausible hypothesis of localized exposure in or around the mill rather than widespread community transmission.

Our results indicate that indoor areas of the mill may have served as sites of exposure or dispersion of spores. Prior workplace-associated blastomycosis outbreaks have primarily occurred in outdoor environments involving direct soil exposure through activities such as excavation, forestry, or construction [24–26]. In contrast, this outbreak occurred in a predominantly indoor industrial facility, where typical environmental reservoirs were not visibly disturbed and exposure pathways were more difficult to identify. Several areas of the mill with increased blastomycosis prevalence, particularly paper machine line #1 and adjacent maintenance areas, had high frequencies of reported indoor pooling water, visible mold, and indoor wind gusts. Although we did not detect *Blastomyces* in these areas, they may reflect environments where spores could have persisted or become aerosolized after being introduced from outside. Notably, large bay doors were frequently open to allow the movement of paper rolls and materials in and out of the mill, and workers often were stationed or moved near these openings. Cross-ventilation was also reported due to open access points and underperforming HVAC systems. These factors likely facilitated the entrainment and recirculation of outdoor air, and potentially spores, into the indoor workspace. However, these findings do not necessarily mean that *Blastomyces* was growing indoors. Rather, it is more likely that spores originating from the outdoor environment entered the mill and that indoor conditions may have contributed to workers' exposure once the spores were present.

Despite testing of 533 indoor and outdoor environmental samples, we did not detect *Blastomyces* using either PCR or culture methods. Growth of environmental fungi in culture samples and detection of fungal DNA by PCR demonstrated the samples were viable. Although *Blastomyces* has been successfully isolated from soil in past outbreaks [22,24,27], environmental detection remains challenging due to the organism's low prevalence, uneven distribution, and the limited quantity of *Blastomyces* present in environmental samples. It is possible that spores were present but below the detection limit or not captured in the sampled areas, particularly given the large size and complex layout of the mill, which made it impractical to sample all potentially contaminated locations. Our findings reinforce the challenge of identifying the environmental reservoir in real-world outbreak settings and highlights the need for continued research on the ecology and environmental persistence of *Blastomyces*, as well as the development of more sensitive and reliable environmental detection methods [13].

Indoor pooling water, visible mold, and wind gusts were all associated with increased prevalence of disease. Although *Blastomyces* is not typically found in standing water, these features may reflect environmental proxies for fungal contamination or spore distribution. Water accumulation may signal damp conditions supportive of fungal persistence, and indoor wind gusts may have facilitated airborne spore movement. The association between reported visible mold and disease should be interpreted cautiously, as spores are microscopic [28]. Workers' reports of visible mold likely reflect other fungi or moisture-related conditions that may contribute to overall indoor environmental quality but do not confirm the presence of *Blastomyces*. Further, such observations are subject to recall bias and personal perception.

Our investigation did not identify any specific associations between exposure to raw wood materials and disease risk. Since *Blastomyces* is thought to grow in decaying wood in the environment [6], we initially hypothesized that spores may have been introduced into the mill on wood or wood chips contaminated with the fungus; however, this pathway remains speculative, and we are not aware of published studies confirming it, particularly in industrial settings. Future research could explore whether raw wood materials serve as potential carriers of *Blastomyces* in endemic regions. Moreover, the papermaking process involves chemical treatments and high temperatures, which makes the survival of fungal spores unlikely. We also found no association between specific papermaking processes and disease risk, suggesting that exposure was more likely related to shared environmental conditions or movement through certain areas of the mill. This interpretation is further supported by the occurrence of cases among administrative workers—who were not involved in production—and the higher case prevalence in mill areas closest to the river.

This investigation had several limitations. The survey and urine antigen testing represent a single point in time, and prevalence estimates may not reflect the full extent of the outbreak. Some cases may have occurred before or after our investigation window, or among workers who did not participate. Although urine antigen testing is sensitive and identified

additional cases [15], the timing of sample collection may have led to missed cases—both among persons with low antigen levels and those who had already recovered. Additionally, exposure locations were self-reported and subject to recall bias, particularly regarding intermittent exposures or subjective environmental conditions such as mold or air movement. We cannot determine with certainty the timing or precise location of exposure. Although we observed temporal clustering of illness onset, exposure likely occurred over a period of weeks to months rather than at a single point in time. Finally, our regression models assumed static work environments, but workers often moved throughout the facility, increasing the challenge of exposure attribution. Furthermore, environmental sampling occurred after many cases had already developed, and we cannot rule out the possibility that routine cleaning or outbreak-related mitigation efforts reduced the presence or detectability of *Blastomyces* in environmental samples. Because no serologic or skin testing was conducted, asymptomatic or subclinical infections may have gone undetected. Case identification was primarily limited to individuals with reported symptoms, medical diagnoses, or those who participated in the medical survey, potentially resulting in underestimation of the true number of infected individuals.

This outbreak highlights the potential for occupational exposure to endemic fungi in indoor work environments, particularly in regions with natural reservoirs of *Blastomyces*. Facilities that draw in outdoor air or store organic material may be at heightened risk. Although environmental conditions like pooling water and poor ventilation do not confirm the presence of *Blastomyces*, they may indicate settings where spores introduced from outside could persist or be redistributed. Following outbreak recognition, we provided interim recommendations during the peak of the epidemic curve. These included improving ventilation system maintenance and filtration, reducing pooling water, minimizing dust disturbance, and using respiratory protection during high-risk activities [29]. Education about blastomycosis symptoms and early medical evaluation for exposed workers was also emphasized. Although a specific environmental source was not identified, these precautionary measures may help reduce future risk. Future work could focus on improving *Blastomyces* environmental detection methods and advancing our understanding of how *Blastomyces* spores are introduced into, transported within, and persist in built environments. Enhanced occupational surveillance and early outbreak recognition remain critical for mitigating the impacts of endemic mycoses in the workplace.

## Disclaimer

The findings and conclusions in this report are those of the author(s) and do not necessarily represent the official position of the NIOSH or CDC. Mention of any company or product does not constitute endorsement by NIOSH/CDC.

## Supporting information

**S1 Fig. Directed acyclic graph (DAG).** DAG, made in DAGitty v3.1, of proposed causal relationships between work locations and environmental conditions at the mill and blastomycosis. *Blastomyces* exposure was unobserved. For associations between work location and blastomycosis, the minimum sufficient adjustment variables were sex and mill tenure. For associations between environmental conditions and blastomycosis, the minimum sufficient adjustment variables were sex, mill tenure, and work location.
(PDF)

**S2 Fig. Prevalence of blastomycosis by primary work location.** Primary work location at the mill was reported by workers from October 2022–April 2023.
(PDF)

**S1 Table. Associations between primary work location at the mill and blastomycosis.**
(PDF)

**S3 Fig. Number of workers with blastomycosis, by illness onset date.** Blastomycosis illness onset dates were plotted by the primary work location of workers' current or most recent job. Illness onset dates were available for 112 mill workers

who participated in the NIOSH medical survey. The average blastomycosis illness onset date was March 4, 2023. Counts less than five were not reported to avoid identification of workers.
(PDF)

**S4 Fig. Primary location of daily exposure to pooling water indoors for mill workers.**
(PDF)

**S5 Fig. Primary location of daily exposure to visible mold indoors for mill workers.**
(PDF)

**S6 Fig. Primary location of daily exposure to wind indoors for mill workers.**
(PDF)

**S2 Table. Environmental samples and PCR/fungal culture results for *Blastomyces* and other fungi.**
(PDF)

## Acknowledgments

Michigan Blastomycosis Outbreak Investigation Team:

Team lead: David N. Weissman, MD, Respiratory Health Division, National Institute for Occupational Safety and Health, Centers for Disease Control and Prevention, Morgantown, West Virginia; email dqw4@cdc.gov. Other team members: Michigan Department of Health and Human Services, Lansing, Michigan: Sara Palmer, MPH; Jevon McFadden, MD; Melissa Millerick-May, PhD; Field Studies Branch, Respiratory Health Division, National Institute for Occupational Safety and Health, Centers for Disease Control and Prevention, Morgantown, West Virginia: Anne Foreman, PhD; Suzanne Tomasi, DVM, MPH; Ethan Fechter-Leggett, DVM, MPVM; Eun Gyung (Emily) Lee, PhD, CIH; Stephen B. Martin, PhD, PE; David N. Weissman, MD; Mycotic Diseases Branch, Centers for Disease Control and Prevention, Atlanta, Georgia, USA: Tom Chiller, PhD; Ana P. Litvintseva, PhD.

The authors also thank mill management, workers, and unions for their participation in the investigation and medical survey and Public Health Delta & Menominee and Michigan Department of Health and Human Services for sharing data. We are grateful to Dr. Gerald Hobbs for his valuable statistical advice and Chengetayi Rimayi for assistance with manuscript formatting. We also thank Caroline Growth and Jeremy Gold for reviewing this manuscript.

## Author contributions

**Conceptualization:** Allyson W. O'Connor, Marcia L. Stanton, Perri C. Callaway, Rachel L. Bailey, Marie A. de Perio, Stella Hines.

**Data curation:** Allyson W. O'Connor, Ju-Hyeong Park, Marcia L. Stanton, Xiaoming Liang, Perri C. Callaway, Ian Hennessee, Mitsuru Toda, Alana Sterkel, Suzanne Dargle, Jean Cox-Ganser.

**Formal analysis:** Allyson W. O'Connor, Xiaoming Liang.

**Funding acquisition:** R. Reid Harvey, Rachel L. Bailey, Marie A. de Perio, Stella Hines, Jean Cox-Ganser.

**Investigation:** Allyson W. O'Connor, Dallas Shi, Ju-Hyeong Park, Marcia L. Stanton, Perri C. Callaway, R. Reid Harvey, Rachel L. Bailey, Ian Hennessee, Mitsuru Toda, Jennifer Meece, Alana Sterkel, Olivia Bree, Jeremy Olstadt, Rebecca Reik, Mary Grace Stobierski, Michael Snyder, Robert Yin, Marie A. de Perio.

**Methodology:** Allyson W. O'Connor, Ju-Hyeong Park, Marcia L. Stanton, Perri C. Callaway, R. Reid Harvey, Rachel L. Bailey, Ian Hennessee, Mitsuru Toda, Jennifer Meece, Alana Sterkel, Suzanne Dargle, Olivia Bree, Jeremy Olstadt, Rebecca Reik, Mary Grace Stobierski, Marie A. de Perio, Jean Cox-Ganser.

**Project administration:** Allyson W. O'Connor, Dallas Shi, Marcia L. Stanton, R. Reid Harvey, Ryan LeBouf, Rachel L. Bailey, Mitsuru Toda, Stella Hines, Jean Cox-Ganser.

**Software:** Xiaoming Liang.

**Supervision:** Marcia L. Stanton, R. Reid Harvey, Ryan LeBouf, Rachel L. Bailey, Mitsuru Toda, Marie A. de Perio, Stella Hines, Jean Cox-Ganser.

**Validation:** Allyson W. O'Connor, Ju-Hyeong Park, Xiaoming Liang, Perri C. Callaway, R. Reid Harvey, Ian Hennessee.

**Visualization:** Allyson W. O'Connor, Dallas Shi, Xiaoming Liang.

**Writing – original draft:** Allyson W. O'Connor, Dallas Shi.

**Writing – review & editing:** Allyson W. O'Connor, Dallas Shi, Ju-Hyeong Park, Marcia L. Stanton, Xiaoming Liang, Perri C. Callaway, R. Reid Harvey, Ryan LeBouf, Rachel L. Bailey, Ian Hennessee, Mitsuru Toda, Jennifer Meece, Alana Sterkel, Olivia Bree, Jeremy Olstadt, Rebecca Reik, Mary Grace Stobierski, Michael Snyder, Robert Yin, Marie A. de Perio, Stella Hines, Jean Cox-Ganser.

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
