## [Decision Letter · Decision Letter 0]

15 Jul 2025

PONE-D-25-24487Assessment of environmental risk factors for blastomycosis during a large outbreak at a Michigan paper millPLOS ONE

Dear Dr. Shi,

Thank you for submitting your manuscript to PLOS ONE. After careful consideration, we feel that it has merit but does not fully meet PLOS ONE’s publication criteria as it currently stands. Therefore, we invite you to submit a revised version of the manuscript that addresses the points raised during the review process.

We look forward to receiving your revised manuscript.

Kind regards,

Rajeev Singh

Academic Editor

PLOS ONE

Journal Requirements:

3. Please amend the manuscript submission data (via Edit Submission) to include author Suzanne Dargle.

4. One of the noted authors is a group or consortium “Michigan Blastomycosis Outbreak Investigation Team”. In addition to naming the author group, please list the individual authors and affiliations within this group in the acknowledgments section of your manuscript. Please also indicate clearly a lead author for this group along with a contact email address.

5. Please upload a copy of Figure 2 to which you refer in your text on page 8. If the figure is no longer to be included as part of the submission please remove all reference to it within the text.

6. Please include your tables as part of your main manuscript and remove the individual files. Please note that supplementary tables (should remain/ be uploaded) as separate "supporting information" files.

8.If the reviewer comments include a recommendation to cite specific previously published works, please review and evaluate these publications to determine whether they are relevant and should be cited. There is no requirement to cite these works unless the editor has indicated otherwise. 

Reviewers' comments:

Reviewer's Responses to Questions

**Comments to the Author**

1. Is the manuscript technically sound, and do the data support the conclusions?

Reviewer #1: Yes

Reviewer #2: Yes

Reviewer #3: Yes

2. Has the statistical analysis been performed appropriately and rigorously? 

Reviewer #1: Yes

Reviewer #2: Yes

Reviewer #3: Yes

3. Have the authors made all data underlying the findings in their manuscript fully available?

Reviewer #1: Yes

Reviewer #2: No

Reviewer #3: Yes

4. Is the manuscript presented in an intelligible fashion and written in standard English?

Reviewer #1: Yes

Reviewer #2: No

Reviewer #3: Yes

5. Review Comments to the Author

Reviewer #1: This manuscript addresses the largest known outbreak of blastomycosis in an indoor industrial setting, which is a highly novel contribution. The implications for occupational health and environmental monitoring are significant.

While the authors rightly emphasize the uncertainty surrounding the exact exposure location, some statements (e.g., “exposure likely occurred within the indoor workplace”) could be softened to better reflect the limitations of retrospective exposure assessment. Consider framing these as "plausible hypotheses supported by circumstantial evidence" rather than near-conclusions.

Some speculation (e.g., spores on raw wood) is plausible but unsupported. Either cite relevant literature or clearly indicate these as hypotheses for future study.

There is a lack of figures/data visualization in this manuscript. Since a sizeable dataset has been collected in this study, it should be utilized effectively to present the key results in a more engaging way.

Reviewer #2: The authors report the results of an epidemiologic investigation into a blastomycosis infection at a paper mill in the UP of Michigan. The methodology is solid. The writing is not.

Comments:

1. Please review the manuscript for scattered errors of English grammar, note that “which” should be preceded by a comma unless preceded by “that” (two instances.} Please use “whereas” instead of “while” unless a temporal connotation is intended (numerous locations.)

2. Please ensure that all abbreviations are defined at first usage. The redefinitions in the disclaimer at the end of the manuscript are redundant.

3. Please provided the manufacturer, city, and state for the statistical and graphics software packages.

4. Please describe how, if the CDC IRB deemed this to be “not research,” all subjects were required to sign informed consent. Would not oral consent have been sufficient? Please provide the accession number for the IRB review.

5. Acronyms referring to genes should be italicized.

6. The authors use “CDC” inconsistently. Please make consistent usage after initial definition of all acronyms (e,g., the acknowledgements.)

7. It is redundant to say that spores are “microscopic” and “cannot be seen with the naked eye.” Please modify.

8. References do not comport with journal style. Please do not capitalize article titles. Use standard journal abbreviations. Use “et al.” after the requisite number of listed authors. Consult the instruction for authors of the journal. Provide the accession date for all website citations. Provide the city of the publisher for the citations of books. List authors initials consistently. Repetitive citations of the statistical software are not needed. Delete, and renumber the references.

9. Hyphenate “non-case” in Tables 1 and 4.

Reviewer #3: "Blastomycosis during a large outbreak at a Michigan paper mill, 2023: A descriptive epidemiologic report"

This paper provides discussed the largest outbreak of blastomycosis in the United States, happening at a paper mill in Michigan. The paper is well-written, however comments below need to be addressed:

1. The information on the of environmental sampling needs further clarification, including the sampling strategy (locations, frequency, sample types) and the rationale for focusing on certain environmental zones such as wood chip areas, ventilation systems).

2. “The interviewer-administered questionnaire collected self-reported data on work history and locations, environmental conditions, medical history, health symptoms, recent respiratory illnesses, blastomycosis-related information, and non-work-related activities…” – this sentence is confusing, is interviewer-administered or self-reported data collected by the authors?

3. The authors do not report whether serologic or skin tests (if available) were used to identify asymptomatic infection. There may be under reporting of mild or asymptomatic cases.

4. The authors do not report on the occupational hygiene assessments such as ventilation conditions and the use of PPE.

5. Are there any recent construction or maintenance activities during the outbreak?

6. The discussion does not sufficiently describe on how the findings are compared to prior blastomycosis outbreaks in the workplace settings.

6. PLOS authors have the option to publish the peer review history of their article (what does this mean? ). If published, this will include your full peer review and any attached files.

**Do you want your identity to be public for this peer review?** For information about this choice, including consent withdrawal, please see our Privacy Policy .

Reviewer #1: No

Reviewer #2: No

Reviewer #3: **Yes: ** YIN-CHENG LIM

---

## [Author Response · Author response to Decision Letter 1]

1 Aug 2025

We appreciate the time and effort by the reviewers on this manuscript. Please see our responses to the comments with line numbers reflecting updated manuscript without tracked changes on.

Reviewer #1: This manuscript addresses the largest known outbreak of blastomycosis in an indoor industrial setting, which is a highly novel contribution. The implications for occupational health and environmental monitoring are significant.

1. While the authors rightly emphasize the uncertainty surrounding the exact exposure location, some statements (e.g., “exposure likely occurred within the indoor workplace”) could be softened to better reflect the limitations of retrospective exposure assessment. Consider framing these as "plausible hypotheses supported by circumstantial evidence" rather than near-conclusions.

Response:

Thank you for this helpful suggestion. We agree that retrospective exposure assessment carries inherent limitations, and we have revised the language throughout the manuscript to more cautiously frame potential exposure scenarios.

Specifically, in the Discussion, we revised:

• Original: “...suggest that exposure may have occurred within the indoor workplace.”

Revised, line 353: “...support the possibility of exposure occurring within the indoor workplace, although we cannot determine this with certainty.”

• Original: “Together, the epidemiologic and genomic findings point to localized exposure in or around the mill rather than widespread community transmission.”

Revised, line 354: “Together, the epidemiologic and genomic findings are consistent with a plausible hypothesis of localized exposure in or around the mill rather than widespread community transmission

2. Some speculation (e.g., spores on raw wood) is plausible but unsupported. Either cite relevant literature or clearly indicate these as hypotheses for future study.

Response:

Thank you for pointing this out. We agree that the possibility of spores being introduced on raw wood remains speculative without supporting data. We have revised the relevant section of the Discussion to explicitly label this as a hypothesis for future investigation and have avoided implying any confirmed role of raw wood in exposure pathways. At this time, we are not aware of published studies directly supporting Blastomyces transmission via raw wood materials in occupational settings, and we have noted this gap.

Specifically, in the Discussion, we revised:

• Original: “It is plausible that spores could have been introduced to the mill on wood that had contacted contaminated soil...”

Revised, line 397: “Since Blastomyces is thought to grow in decaying wood in the environment (6), we initially hypothesized that spores may have been introduced into the mill on wood or wood chips contaminated with the fungus; however, this pathway remains speculative, and we are not aware of published studies confirming it, particularly in industrial settings. Future research could explore whether raw wood materials serve as potential carriers of Blastomyces in endemic regions.”

3. There is a lack of figures/data visualization in this manuscript. Since a sizeable dataset has been collected in this study, it should be utilized effectively to present the key results in a more engaging way.

Response:

Thank you for this thoughtful suggestion. We agree that visual data presentation can improve clarity and reader engagement. In response to your feedback, we have revised the manuscript to enhance data visualization.

We have replaced the original Tables 2, 3, 5, and 6 with figures of forest plots to more effectively convey the adjusted prevalence ratios and confidence intervals associated with key workplace exposures. These updated figures provide a clearer and more intuitive depiction of the statistical findings. We believe this change improves the visual presentation of the data and better supports reader interpretation.

Additionally, we have retained Figure 1 in the main text and included several supplemental figures (Figures A2–A6), which illustrate other important findings, such as the temporal distribution of cases and environmental exposure reports. We have ensured that all figures are clearly referenced in the main text and are accompanied by descriptive captions.

We appreciate the reviewer’s recommendation and believe these revisions strengthen the manuscript’s accessibility and impact.

Reviewer #2: The authors report the results of an epidemiologic investigation into a blastomycosis infection at a paper mill in the UP of Michigan. The methodology is solid. The writing is not.

Comments:

1. Please review the manuscript for scattered errors of English grammar, note that “which” should be preceded by a comma unless preceded by “that” (two instances.} Please use “whereas” instead of “while” unless a temporal connotation is intended (numerous locations.)

Response:

Thank you for this detailed editorial feedback. We have carefully reviewed the manuscript and corrected instances where “which” was not properly preceded by a comma when used in non-restrictive clauses. We also replaced instances of “while” with “whereas” in comparative or contrastive statements that do not convey a temporal meaning. These edits have been incorporated throughout the revised manuscript to improve clarity and consistency.

2. Please ensure that all abbreviations are defined at first usage. The redefinitions in the disclaimer at the end of the manuscript are redundant.

Response:

Thank you for this helpful suggestion. We reviewed the manuscript to ensure that all abbreviations are defined at first use in the main text, tables, and figure captions. We have removed redundant redefinitions in the disclaimer section to avoid unnecessary repetition and improve clarity. All abbreviations now follow standard conventions as recommended by the journal.

3. Please provided the manufacturer, city, and state for the statistical and graphics software packages.

Response:

Thank you for this reminder. We have updated the Methods section to include the full manufacturer information for all statistical and graphics software packages used in the analysis.

In the Methods > Data analysis section, we revised the sentence:

• Original: “Analyses were conducted using SAS® software version 9.4 and JMP® 16.1.”

Revised, line 183: “Analyses were conducted using SAS® software version 9.4 (SAS Institute Inc., Cary, NC), JMP® version 16.1 (SAS Institute Inc., Cary, NC) [19], and R (version 4.5.1; https://www.r-project.org/). ”

• Original: “DAGs were created using DAGitty version 3.1.”

Revised, line 185: “DAGs were created using DAGitty version 3.1 (www.dagitty.net; developed by Johannes Textor, Radboud University Medical Center, Nijmegen, The Netherlands)

4. Please describe how, if the CDC IRB deemed this to be “not research,” all subjects were required to sign informed consent. Would not oral consent have been sufficient? Please provide the accession number for the IRB review.

Response: We appreciate the opportunity to clarify. This activity was reviewed by CDC and determined to be public health practice, not research, and therefore not subject to institutional review board (IRB) oversight. It was conducted as part of the NIOSH Health Hazard Evaluation Program. Nevertheless, written informed consent was obtained from all participants to ensure transparency and to protect participant autonomy, as the medical survey included urine antigen testing and collection of potentially sensitive health information. The consent process described the voluntary nature of participation, the confidentiality of collected data, and the minimal risks and benefits involved. We have updated the manuscript to include this clarification and the applicable regulatory citations. As the activity was determined to be non-research, no IRB accession number applies.

Changes Made:

In the Ethical review and participant consent section, we revised the paragraph on line 239 to:

This activity was reviewed by CDC and conducted in compliance with applicable federal law and CDC policy (45 C.F.R. part 46.102(l)(2); 21 C.F.R. part 56; 42 U.S.C. §241(d); 5 U.S.C. §552a; 44 U.S.C. §3501 et seq.). The study was determined not to be human subjects research but to be public health practice as part of the NIOSH Health Hazard Evaluation Program and thus was not subject to institutional review board review. All participants provided written informed consent before participation in the medical survey. The informed consent included information about the purpose and duration of the medical survey, what participants would be asked to do, and the benefits and minimal risks involved, as well as that personal information collected was confidential and would be protected to the extent allowed by law, participation was voluntary, and participants could withdraw from the evaluation at any time, for any reason.

5. Acronyms referring to genes should be italicized.

Response:

Thank you for this important clarification. We have reviewed the manuscript and updated the formatting to italicize all gene names and acronyms, including BAD-1 and ITS3/4, in accordance with standard scientific conventions for gene nomenclature. Protein products (if mentioned) remain in regular (non-italicized) font as appropriate.

6. The authors use “CDC” inconsistently. Please make consistent usage after initial definition of all acronyms (e,g., the acknowledgements.)

Response:

Thank you for noting this inconsistency. We have reviewed the manuscript and supplemental materials to ensure consistent use of “CDC” after its initial definition as “Centers for Disease Control and Prevention.” All subsequent mentions now use the acronym “CDC,” including in the acknowledgments section and author affiliations. We also verified consistent usage for related acronyms such as “NIOSH.”

7. It is redundant to say that spores are “microscopic” and “cannot be seen with the naked eye.” Please modify.

Response:

Thank you for pointing this out. We have revised the sentence to eliminate redundancy while preserving the intended meaning. In line 391, The updated text refers only to the microscopic nature of spores, which is sufficient to convey that they are not visible without magnification: “...as Blastomyces spores are microscopic.”

8. References do not comport with journal style. Please do not capitalize article titles. Use standard journal abbreviations. Use “et al.” after the requisite number of listed authors. Consult the instruction for authors of the journal. Provide the accession date for all website citations. Provide the city of the publisher for the citations of books. List authors initials consistently. Repetitive citations of the statistical software are not needed. Delete, and renumber the references.

Response:

Thank you for this detailed feedback. We have carefully revised the reference list to comply with PLOS ONE formatting requirements. Specifically:

• Article titles are now in sentence case (only the first word and proper nouns capitalized).

• Journal names are abbreviated according to standard scientific indexing conventions (e.g., PubMed).

• Author lists follow the appropriate truncation using “et al.” after six authors, as specified in the journal’s author guidelines.

• Authors’ names and initials are now listed consistently.

• Accession dates have been added for all website citations.

• City and publisher information has been added for all book citations.

• Duplicate references to software (e.g., SAS, JMP) have been consolidated and removed.

• All references have been renumbered accordingly in both the reference list and in-text citations.

9. Hyphenate “non-case” in Tables 1 and 4.

Response:

Thank you for catching this formatting inconsistency. We have hyphenated “non-case” in Tables 1 and 2 (Table number revised from Table 4).

Reviewer #3: "Blastomycosis during a large outbreak at a Michigan paper mill, 2023: A descriptive epidemiologic report"

This paper provides discussed the largest outbreak of blastomycosis in the United States, happening at a paper mill in Michigan. The paper is well-written, however comments below need to be addressed:

1. The information on the of environmental sampling needs further clarification, including the sampling strategy (locations, frequency, sample types) and the rationale for focusing on certain environmental zones such as wood chip areas, ventilation systems).

Response:

Thank you for this helpful suggestion. We have revised the Environmental sampling and analyses section to provide additional detail on the rationale and design of our sampling strategy.

Specifically, we added or revised text in the Methods section as follows:

• Original: “Sampling was conducted during the initial site visit and again later in the summer to capture seasonal variation…”

Revised, line 191: “We conducted the main indoor and outdoor sampling during the medical survey period. An additional outdoor sampling occurred in summer, coinciding with the construction of a bridge within the mill property. Indoor and outdoor sampling locations were selected based on three key criteria: the presence of indoor blastomycosis cases, their proximity to suspected environmental reservoirs (e.g., riverbanks, ponds, wood chip piles), and evidence of water intrusion.”

• New text added on line 196: “We prioritized the wood chip storage areas due to their high organic content and contact with outdoor soil, which may support fungal growth. Indoors, HVAC filters, duct liners, and accumulated surface dust were targeted to assess whether spores may have been entrained, redistributed through ventilation, or settled in shared workspaces. A total of 533 environmental samples were collected, including outdoor soils (n = 113), raw wood materials (n = 9), indoor surface dust (n = 344), and HVAC filters, duct liners, and pooled water (n = 67). These sample types reflect potential environmental reservoirs or transport pathways for Blastomyces, with an emphasis on both entry from the outdoor environment and possible persistence or movement within indoor work areas.”

2. “The interviewer-administered questionnaire collected self-reported data on work history and locations, environmental conditions, medical history, health symptoms, recent respiratory illnesses, blastomycosis-related information, and non-work-related activities…” – this sentence is confusing, is interviewer-administered or self-reported data collected by the authors?

Response:

Thank you for pointing out this ambiguity. We clarified the language to distinguish between the mode of data collection and the source of the information. The questionnaire was administered by trained interviewers, but the responses were self-reported by participants. We have revised the sentence accordingly for clarity:

• Original: “The interviewer-administered questionnaire collected self-reported data on work history and locations, environmental conditions, medical history, health symptoms, recent respiratory illnesses, blastomycosis-related information, and non-work-related activities…”

• Revised, line 120: “Trained interviewers administered a structured questionnaire that captured participants’ self-reported information on work history and locations, environmental conditions, medical history, health symptoms, recent respiratory illnesses, blastomycosis-related experiences, and non-work-related activities…”

3. The authors do not report whether serologic or skin tests (if available) were used to identify asymptomatic infection. There may be under reporting of mild or asymptomatic cases.

Response:

We appreciate this important observation and have clarified in both the Methods and Discussion sections that no serologic or skin testing was conducted as part of this investigation. While these methods have historically been used in some blastomycosis outbreak assessments, skin testing is no longer widely available, and most serologic assays suffer from low sensitivity and limited utility in distinguishing recent from prior infections in endemic areas. A newly approved enzyme immunoassay does offer improved sensitivity; however, it remains unclear how long antibodies persist post-infection, especially in populations with prior environm

---

## [Editor Report · Decision Letter 1]

31 Aug 2025

Assessment of environmental risk factors for blastomycosis during a large outbreak at a Michigan paper mill

PONE-D-25-24487R1

Dear Dr. Shi,

We’re pleased to inform you that your manuscript has been judged scientifically suitable for publication and will be formally accepted for publication once it meets all outstanding technical requirements.

Kind regards,

Rajeev Singh

Academic Editor

PLOS ONE
---

## [Editor Report · Acceptance letter]

PONE-D-25-24487R1

PLOS ONE

Dear Dr. Shi,

I'm pleased to inform you that your manuscript has been deemed suitable for publication in PLOS ONE. Congratulations! Your manuscript is now being handed over to our production team.

Kind regards,

on behalf of

Dr. Rajeev Singh

Academic Editor

PLOS ONE